# Learning convex bounds for linear quadratic control policy synthesis

**Jack Umenberger**
Department of Information Technology
Uppsala University
Sweden
jack.umenberger@it.uu.se

**Thomas B. Schön**
Department of Information Technology
Uppsala University
Sweden
thomas.schon@it.uu.se

## Abstract

Learning to make decisions from observed data in dynamic environments remains a problem of fundamental importance in a number of fields, from artificial intelligence and robotics, to medicine and finance. This paper concerns the problem of learning control policies for unknown linear dynamical systems so as to maximize a quadratic reward function. We present a method to optimize the expected value of the reward over the posterior distribution of the unknown system parameters, given data. The algorithm involves sequential convex programing, and enjoys reliable local convergence and robust stability guarantees. Numerical simulations and stabilization of a real-world inverted pendulum are used to demonstrate the approach, with strong performance and robustness properties observed in both.

## 1   Introduction

Decision making for dynamical systems in the presence of uncertainty is a problem of great prevalence and importance, as well as considerable difficulty, especially when knowledge of the dynamics is available only via limited observations of system behavior. In machine learning, the data-driven search for a control policy to maximize the expected reward attained by a stochastic dynamic process is known as *reinforcement learning* (RL) [45]. Despite remarkable recent success in games [32, 43], a major obstacle to the deployment RL-based control on physical systems (e.g. robots and self-driving cars) is the issue of *robustness*, i.e., guaranteed safe and reliable operation. With the necessity of such guarantees widely acknowledged [2], so-called 'safe RL' remains an active area of research [21].

The problem of robust automatic decision making for uncertain dynamical systems has also been the subject of intense study in the area of *robust control* (RC) [57]. In RC, one works with a set of plausible models and seeks a control policy that is guaranteed to stabilize all models within the set. In addition, there is also a performance objective to optimize, i.e. a reward to be maximized, or equivalently, a cost to be minimized. Such cost functions are usually defined with reference to either a nominal model [20, 25] or the worst-case model [36] in the set. RC has been extremely successful in a number of engineering applications [38]; however, as has been noted, e.g., [48, 35], robustness may (understandably) come at the expense of performance, particularly for worst-case design.

The problem we address in this paper lies at the intersection of reinforcement learning and robust control, and can be summarized as follows: given observations from an unknown dynamical system, we seek a policy to optimize the expected cost (as in RL), subject to certain robust stability guarantees (as in RC). Specifically, we focus our attention on control of linear time-invariant dynamical systems, subject to Gaussian disturbances, with the goal of minimizing a quadratic function penalizing state deviations and control action. When the system is known, this is the classical linear quadratic regulator (LQR), a.k.a. $H_2$, optimal control problem [8]. We are interested in the setting in which the system is unknown, and knowledge of the dynamics must be inferred from observed data.

**Contributions and paper structure** The principal contribution of this paper is an algorithm to optimize the expected value of the linear quadratic regulator reward/cost function, where the expectation is w.r.t. the posterior distribution of unknown system parameters, given observed data; cf. Section 3 for a detailed problem formulation. Specifically, we construct a sequence of convex approximations (upper bounds) to the expected cost, that can be optimized via semidefinite programing [50]. The algorithm, developed in Section 4, invokes the majorize-minimization (MM) principle [29], and consequently enjoys reliable convergence to local optima. An important part of our contribution lies in guarantees on the robust stability properties of the resulting control policies, cf. Section 4.3. We demonstrate the proposed method via two experimental case studies: i) the benchmark problem on simulated systems considered in [17, 48], and ii) stabilization of a real-world inverted pendulum. Strong performance and robustness properties are observed in both. Moving forward, from a machine learning perspective this work contributes to the growing body of research concerned with ensuring robustness in RL, cf. Section 2. From a control perspective, this work appropriates cost functions more commonly found in RL (namely, expected reward) to a RC setting, with the objective of reducing conservatism of the resulting robust control policies.

## 2 Related work

Incorporating various notions of 'robustness' into RL has long been an area of active research [21]. In so-called 'safe RL', one seeks to respect certain safety constraints during exploration and/or policy optimization, for example, avoiding undesirable regions of the state-action space [22, 1]. A related problem is addressed in 'risk-sensitive RL', in which the search for a policy takes both the expected value and variance of the reward into account [31, 19]. Recently, there has been an increased interest in notions of robustness more commonly considered in control theory, chiefly *stability* [35, 3]. Of particular relevance is the work of [4], which employs Lyapunov theory [27] to verify stability of learned policies. Like the present paper, [4] adopts a Bayesian framework; however, [4] makes use of Gaussian processes [39] to model the uncertain nonlinear dynamics, which are assumed to be deterministic. A major difference between [4] and our work is the cost function; in the former the policy is selected by optimizing for worst-case performance, whereas we optimize the expected cost.

Robustness of data-driven control has also been the focus of a recently developed family of methods referred to as 'coarse-ID control', cf. [47, 17, 7, 44], in which finite-data bounds on the accuracy of the least squares estimator are combined with modern robust control tools, such as *system level synthesis* [55]. Coarse-ID builds upon so-called '$H_\infty$ identification' methods for learning models of dynamical systems, along with error bounds that are compatible with robust synthesis methods [26, 14, 13]. $H_\infty$ identification assumes an adversarial (i.e. worst-case) disturbance model, whereas Coarse-ID is applicable to probabilistic models, such as those considered in the present paper. Of particular relevance to the present paper is [17], which provides sample complexity bounds on the performance of robust control synthesis for the infinite horizon LQR problem, when the true system is not known. Such bounds necessarily consider the worst-case model, given the observed data, where as we are concerned with expected cost over the posterior distribution of models.

This approach of controller synthesis w.r.t. distributions over models has much in common with the field of probabilistic robust control [11, 46]. Early work in this area applied statistical learning theory [53] to randomized algorithms for feasibility analysis and policy design, cf. e.g., [51, 52]. Of particular relevance to the present paper is the so-called 'scenario approach' to control: robustness requirements lead to semi-infinite convex programs, which are approximated by sampling a finite number of constraints, cf. e.g., [9, 10]. A key focus of the scenario approach is bounding sample complexity (i.e., the number of sampled constraints required to ensure some probability of feasibility), without resorting to statistical learning theory, so as to reduce conservatism.

In closing, we briefly mention the so-called 'Riemann-Stieltjes' class of optimal control problems, for uncertain continuous-time dynamical systems, cf. e.g., [41, 40]. Such problems often arise in aerospace applications (e.g. satellite control) where the objective is to design an open-loop control signal (e.g. for an orbital maneuver) rather than a feedback policy.

## 3 Problem formulation

In this section we describe in detail the specific problem that we address in this paper. The following notation is used: $\mathbb{S}^n$ denotes the set of $n \times n$ symmetric matrices; $\mathbb{S}^n_+$ ($\mathbb{S}^n_{++}$) denotes the cone of positive semdefinite (positive definite) matrices. $A \succeq B$ denotes $A - B \in \mathbb{S}^n_+$, similarly for $\succ$ and $\mathbb{S}^n_{++}$. The trace of $A$ is denoted $\operatorname{tr} A$. The transpose of $A$ is denoted $A'$. $|a|^2_Q$ is shorthand for $a'Qa$. The convex hull of set $\Theta$ is denoted $\operatorname{conv}\Theta$. The set of Schur stable matrices is denoted $\mathcal{S}$.

**Dynamics, reward function and policies**   We are concerned with control of discrete linear time-invariant dynamical systems of the form

$$x_{t+1} = Ax_t + Bu_t + w_t, \qquad w_t \sim \mathcal{N}(0, \Pi), \tag{1}$$

where $x_t \in \mathbb{R}^{n_x}$, $u_t \in \mathbb{R}^{n_u}$, and $w_t \in \mathbb{R}^{n_w}$ denote the state, input, and unobserved exogenous disturbance at time $t$, respectively. Let $\theta := \{A, B, \Pi\}$. Our objective is to design a feedback control policy $u_t = \phi(x_t)$ that minimizes the cost function $\lim_{T\to\infty} \frac{1}{T} \sum_{t=0}^T \mathbb{E}[x_t'Qx_t + u_t'Ru_t]$, where $x_t$ evolves according to (1), and $Q \succeq 0$ and $R \succ 0$ are user defined weight matrices. A number of different parametrizations of the policy $\phi$ have been considered in the literature, from neural networks (popular in RL, e.g., [4]) to causal (typically linear) dynamical systems (common in RC, e.g., [36]). In this paper, we will restrict our attention to static-gain policies of the form $u_t = Kx_t$, where $K \in \mathbb{R}^{n_u \times n_x}$ is constant. As noted in [17], controller synthesis and implementation, is simpler (and more computationally efficient) for such policies. When the parameters of the true system, denoted $\theta_{\text{tr}} := \{A_{\text{tr}}, B_{\text{tr}}, \Pi_{\text{tr}}\}$, are known this is the infinite horizon LQR problem, the optimal solution of which is well-known [5]. We assume that $\theta_{\text{tr}}$ is unknown; rather, our knowledge of the dynamics must be inferred from observed sequences of inputs and states.

**Observed data**   We adopt the data-driven setup used in [17], and assume that $\mathcal{D} := \{x_{0:T}^r, u_{0:T}^r\}_{r=1}^N$ where $x_{0:T}^r = \{x_t^r\}_{t=0}^T$ is the observed state sequence attained by evolving the true system for $T$ time steps, starting from an arbitrary $x_0^r$ and driven by arbitrary input $u_{0:T}^r = \{u_t^r\}_{t=0}^T$. Each of these $N$ independent experiments is referred to as a *rollout*. We perform parameter inference in the offline/batch setting; i.e., all data $\mathcal{D}$ is assumed to be available at the time of controller synthesis.

**Optimization objective**   Given observed data and, possibly, prior knowledge of the system, we then have the posterior distribution over the model parameters denoted $\pi(\theta) := p(A, B, \Pi|\mathcal{D})$, in place of the true parameters $\theta_{\text{tr}}$. The function that we seek to minimize is the expected cost w.r.t. the posterior distribution, i.e.,

$$\lim_{T\to\infty} \frac{1}{T} \sum_{t=0}^T \mathbb{E}[x_t'Qx_t + u_t'Ru_t \mid x_{t+1} = Ax_t + Bu_t + w_t,\ w_t \sim \mathcal{N}(0, \Pi),\ \{A, B, \Pi\} \sim \pi(\theta)]. \tag{2}$$

In practice, the support of $\pi$ almost surely contains $\{A, B\}$ that are unstabilizable, which implies that (2) is infinite. Consequently, we shall consider averages over confidence regions w.r.t. $\pi$. For convenience, let us denote the infinite horizon LQR cost, for given system parameters $\theta$, by

$$J(K|\theta) := \lim_{t\to\infty} \mathbb{E}[x_t'(Q + K'RK)x_t \mid x_{t+1} = (A + BK)x_t + w_t,\ w \sim \mathcal{N}(0, \Pi)] \tag{3a}$$

$$= \begin{cases} \operatorname{tr} X\Pi \text{ with } X = (A + BK)'X(A + BK) + Q + K'RK, & A + BK \in \mathcal{S} \\ \infty, & \text{otherwise,} \end{cases} \tag{3b}$$

where the second equality follows from standard Gramian calculations, and $\mathcal{S}$ denotes the set of Schur stable matrices. As an alternative to (2) we may consider a cost function like $J^c(K) := \int_{\Theta^c} J(K|\theta)\pi(\theta)d\theta$, where $\Theta^c$ denotes a $c$ % confidence region of the parameter space w.r.t. the posterior $\pi$. Though better suited to optimization than (2), which is almost surely infinite, this integral cannot be evaluated in closed form, due to the complexity of $J(\cdot|\theta)$ w.r.t. $\theta$. Furthermore, there is still no guarantee that $\Theta^c$ contain *only* stabilizable models. To circumvent both of these issues, we propose the following Monte Carlo (MC) approximation of $J^c(K)$,

$$J_M^c(K) := \frac{1}{M} \sum_{i=1}^M J(K|\theta_i), \qquad \theta_i \sim \Theta^c \cap \mathcal{M}, \qquad i = 1, \dots, M, \tag{4}$$

where $M$ is the number of samples used, and $\mathcal{M}$ denotes the set of stabilizable $\{A, B\}$. Note that (4) is not a true MC approximation of $J^c(K)$ as only stabilizable samples $\{A_i, B_i\} \in \mathcal{M}$ are used.

**Posterior distribution** Given data $\mathcal{D}$, the parameter posterior distribution is given by Bayes' rule:

$$\pi(\theta) := p(\theta|\mathcal{D}) = \frac{1}{p(\mathcal{D})} p(\mathcal{D}|\theta)p(\theta) \propto p(\theta) \prod_{r=1}^{N} \prod_{t=1}^{T} p(x_t^r|x_{t-1}^r, u_{t-1}^r, \theta) =: \bar{\pi}(\theta), \quad (5)$$

where $p(\theta)$ denotes our prior belief on $\theta$, $p(x_t^r|x_{t-1}^r, u_{t-1}^r, \theta) = \mathcal{N}\left(Ax_{t-1}^r + Bu_{t-1}^r, \Pi\right)$, and $\bar{\pi} = p(\mathcal{D})\pi$ denotes the unnormalized posterior. To sample from $\pi$, we can distinguish between two different cases. First, consider the case when $\Pi_{\text{tr}}$ is known or can be reliably estimated independently of $\{A, B\}$. This is the setting in, e.g., [17]. In this case, the likelihood can be equivalently expressed as a Gaussian distribution over $\{A, B\}$. Then, when the prior $p(A, B)$ is uniform (i.e. non-informative) or Gaussian (self-conjugate), the posterior $p(A, B|\Pi_{\text{tr}}, \mathcal{D})$ is also Gaussian, cf. Appendix A.1.1. Second, consider the general case in which $\Pi_{\text{tr}}$, along with $\{A, B\}$, is unknown. In this setting, one can select from a number of methods adapted for Bayesian inference in dynamical systems, such as Metropolis-Hastings [33], Hamiltonian Monte Carlo [15], and Gibbs sampling [16, 56]. When one places a non-informative prior on $\Pi$ (e.g., $p(\Pi) \propto \det(\Pi)^{-\frac{n_x+1}{2}}$), each iteration of a Gibbs sampler targeting $\pi$ requires sampling from either a Gaussian or an inverse Wishart distribution, for which reliable numerical methods exist; cf. Appendix A.1.2. In both of these cases we can sample from $\pi$ and evaluate $\bar{\pi}$ point-wise. To draw $\theta_i \sim \Theta^c \cap \mathcal{M}$, as in (4), we can first draw a large number of samples from $\pi$, discard the $(100-c)\%$ of samples with the lowest unnormalized posterior values, and then further discard any samples that happen to be unstabilizable. For convenience, we define $\tilde{\Theta}_M^c := \{\{\theta_i\}_{i=1}^M : \theta_i \sim \Theta^c \cap \mathcal{M}, i = 1, \ldots, M\}$, which should be interpreted as a set of $M$ realizations of this procedure for sampling $\theta_i \sim \Theta^c \cap \mathcal{M}$.

**Summary** We seek the solution of the optimization problem $\min_K J_M^c(K)$ for $K \in \mathbb{R}^{n_u \times n_x}$.

# 4 Solution via semidefinite programing

In this section we present the principal contribution of this paper: a method for solving $\min_K J_M^c(K)$ via convex (semidefinite) programing (SDP). It is convenient to consider an equivalent representation

$$\min_{K, \{X_i\}_{i=1}^M \in \mathbb{S}_{++}^{n_x}} \frac{1}{M} \sum_{i=1}^{M} \text{tr } X_i \Pi_i, \tag{6a}$$

$$\text{s.t.} \qquad X_i \succeq (A_i + B_iK)'X_i(A_i + B_iK) + Q + K'RK, \ \{A_i, B_i, \Pi_i\} \in \tilde{\Theta}_M^c, \quad (6b)$$

where the Comparison Lemma [34, Lecture 2] has been used to replace the equality in (3b) with the inequality in (6b). We introduce the notation $\mathbb{S}_\epsilon^n := \{S \in \mathbb{S}^n : S \succeq \epsilon I, S \preceq \mu I\}$, where $\epsilon$ and $\mu$ are arbitrarily small and large positive constants, respectively. $\mathbb{S}_\epsilon^n$ serves as a compact approximation of $\mathbb{S}_{++}^n$, suitable for use with SDP solvers, i.e., $S \in \mathbb{S}_\epsilon^n \implies S \in \mathbb{S}_{++}^n$.

## 4.1 Common Lyapunov relaxation

The principal challenge in solving (6) is that the constraint (6b) is not jointly convex in $K$ and $X^i$. The usual approach to circumventing this nonconvexity is to first apply the Schur complement to (6b), and then conjugate by the matrix $\text{diag}(X_i^{-1}, I, I, I)$, which leads to the equivalent constraint

$$\begin{bmatrix} X_i^{-1} & X_i^{-1}(A_i + B_iK)' & X_i^{-1}Q^{1/2} & X_i^{-1}K' \\ (A_i + B_iK)X_i^{-1} & X_i^{-1} & 0 & 0 \\ Q^{1/2}X_i^{-1} & 0 & I & 0 \\ KX_i^{-1} & 0 & 0 & R^{-1} \end{bmatrix} \succeq 0. \tag{7}$$

With the change of variables $Y_i = X_i^{-1}$ and $L_i = KX_i^{-1}$, (7) becomes an linear matrix inequality (LMI), in $Y_i$ and $L_i$. This approach is effective when $M = 1$ (i.e. we have a single nominal system, as in standard LQR). However, when $M > 1$ we cannot introduce a new $Y_i$ for each $X_i^{-1}$, as we lose uniqueness of the controller $K$ in $L_i = KX_i^{-1}$, i.e., in general $L_iY_i^{-1} \neq L_jY_j^{-1}$ for $i \neq j$. One strategy (prevalent in robust control, e.g., [17, §C]) is to employ a 'common Lyapunov function', i.e., $Y = X_i^{-1}$ for all $i = 1, \ldots, M$. This gives the following convex relaxation (upper bound) of

problem (6),

$$\min_{K,\, Y\in\mathbb{S}^{n_x}_\epsilon,\, \{Z_i\}^M_{i=1}\in\mathbb{S}^{n_x}} \mathrm{tr}\, Z_i, \tag{8a}$$

$$\text{s.t.} \quad \begin{bmatrix} Z^i & G_i \\ G'_i & Y \end{bmatrix} \succeq 0, \quad \begin{bmatrix} Y & YA'_i + L'B'_i & YQ^{1/2} & L' \\ A_iY + B_iL & Y & 0 & 0 \\ Q^{1/2}Y & 0 & I & 0 \\ L & 0 & 0 & R^{-1} \end{bmatrix} \succeq 0,\ \theta_i \in \tilde{\Theta}^c_M, \tag{8b}$$

where $G_i$ denotes the Cholesky factorization of $\Pi_i$, i.e., $\Pi_i = G_iG'_i$, and $\{Z^i\}^M_{i=1}$ are slack variables used to encode the cost (6a) with the change of variables, i.e.,

$$\min_Y \mathrm{tr}\, Y^{-1}\Pi_i \le \big\{ \min_{Y,Z_i} \mathrm{tr}\, Z_i \ \text{ s.t. } \ Z_i \succeq G'_iY^{-1}G_i \big\} \iff \min_{Y,Z_i} \mathrm{tr}\, Z_i \ \text{ s.t. } \ \begin{bmatrix} Z_i & G_i \\ G'_i & Y \end{bmatrix} \succeq 0.$$

The approximation in (8) is highly conservative, which motivates the iterative local optimization method presented in Section 4.2. Nevertheless, (8) provides a principled way (i.e., a one-shot convex program) to initialize the iterative search method derived in Section 4.2.

## 4.2 Iterative improvement by sequential semidefinite programing

To develop this iterative search method first consider an equivalent representation of $J(K|\theta_i)$,

$$J(K|\theta_i) = \min_{X_i\in\mathbb{S}^{n_x}_\epsilon} \mathrm{tr}\ X_i\Pi_i \tag{9a}$$

$$\text{s.t.} \quad \begin{bmatrix} X_i - Q & (A_i+B_iK)' & K' \\ A_i+B_iK & X_i^{-1} & 0 \\ K & 0 & R^{-1} \end{bmatrix} \succeq 0, \quad \text{recall: } \theta_i = \{A_i, B_i, \Pi_i\}. \tag{9b}$$

This representation highlights the nonconvexity of $J(K|\theta_i)$ due to the $X_i^{-1}$ term, which was addressed (in the usual way) by a change of variables in Section 4.1. In this section, we will instead replace $X_i^{-1}$ with a linear approximation and prove that this leads to a tight convex upper bound. Given $S \in \mathbb{S}^n_{++}$, let $T(S, S_0)$ denote the first order (i.e. linear) Taylor series approximation of $S^{-1}$ about some nominal $S_0 \in \mathbb{S}^n_{++}$, i.e., $T(S, S_0) := S_0^{-1} + \frac{\partial S^{-1}}{\partial S}\Big|_{S=S_0} (S - S_0) = S_0^{-1} - S_0^{-1}(S - S_0)S_0^{-1}$. We now define the function

$$\hat{J}(K, \bar{K}|\theta_i) := \min_{X_i\in\mathbb{S}^{n_x}_\epsilon} \mathrm{tr}\ X_i\Pi_i \tag{10a}$$

$$\text{s.t.} \quad \begin{bmatrix} X_i - Q & (A_i+B_iK)' & K' \\ A_i+B_iK & T(X_i,\bar{X}_i) & 0 \\ K & 0 & R^{-1} \end{bmatrix} \succeq 0, \tag{10b}$$

where $\bar{X}_i$ is any $X_i \in \mathbb{S}^{n_x}_\epsilon$ that achieves the minimum in (9), with $K = \bar{K}$ for some nominal $\bar{K}$, i.e., $J(\bar{K}|\theta_i) = \mathrm{tr}\ \bar{X}_i\Pi_i$. Analogously to (4), we define

$$\hat{J}^c_M(K, \bar{K}) := \frac{1}{M}\sum_{\theta_i\in\tilde{\Theta}^c_M} \hat{J}(K, \bar{K}|\theta_i). \tag{11}$$

We now show that $\hat{J}^c_M(K, \bar{K})$ is a convex upper bound on $J^c_M(K)$, which is tight at $K = \bar{K}$. The proof is given in A.2.2 and makes use of the following technical lemma (cf. A.2.1 for proof),

**Lemma 4.1.** $T(S, S_0) \preceq S^{-1}$ for all $S, S_0 \in \mathbb{S}^n_{++}$, where $T(S, S_0)$ denotes the first-order Taylor series expansion of $S^{-1}$ about $S_0$.

**Theorem 4.1.** Let $\hat{J}^c_M(K, \bar{K})$ be defined as in (11), with $\bar{K}$ such that $J^c_M(\bar{K})$ is finite. Then $\hat{J}^c_M(K, \bar{K})$ is a convex upper bound on $J^c_M(K)$, i.e., $\hat{J}^c_M(K, \bar{K}) \ge J^c_M(K)\ \forall K$. Furthermore, the bound is 'tight' at $\bar{K}$, i.e., $\hat{J}^c_M(\bar{K}, \bar{K}) = J^c_M(\bar{K})$.

**Iterative algorithm** To improve upon the common Lyapunov solution given by (8), we can solve a sequence of convex optimization problems: $K^{(k+1)} = \arg\min_K \hat{J}^c_M(K, K^{(k)})$, cf. Algorithm 1 for details. This procedure of optimizing tight surrogate functions in lieu of the actual objective

function is an example of the 'majorize-minimization (MM) principle', a.k.a. optimization transfer [29]. MM algorithms enjoy good numerical robustness, and (with the exception of some pathological cases) reliable convergence to local minima [49]. Indeed, it is readily verified that $J_M^c(K^{(k)}) = \hat{J}_M^c(K^{(k)}, K^{(k)}) \geq \hat{J}_M^c(K^{(k+1)}, K^{(k)}) \geq J_M^c(K^{(k+1)})$, where equality follows from tightness of the bound, and the second inequality is due to the fact that $\hat{J}_M^c(K, K^{(k)})$ is an upper bound. This implies that $\{J_M^c(K^{(k)})\}_{k=1}^\infty$ is a converging sequence.

Before proceeding, let us comment briefly on the computational complexity of the approach, which will be dominated by the convex program $\min_K \hat{J}_M^c(K, \bar{K})$ in (11). The complexity of each iteration of an interior point method for solving this problem is $\mathcal{O}(\max\{m^3, Mmn^3, Mm^2n^2\})$, cf. e.g. [30, §2], where $m = n_x n_u + Mn_x(n_x+1)/2$ denotes the dimensionality of the decision variable, and $n = 2n_x + n_u$ denotes the dimension of the LMI in (10b). It has been observed that the number of iterations required for convergence grows slowly with problem dimension [50]. For computation times on numerical examples, refer to Table 2.

---

**Algorithm 1** Optimization of $J_M^c(K)$ via semidefinite programing

---

1: **Input:** observed data $\mathcal{D}$, confidence $c$, LQR cost matrices $Q$ and $R$, number of particles in Monte Carlo approximation $M$, convergence tolerance $\epsilon$.
2: Generate $M$ samples from $\Theta^c \cap \mathcal{M}$, i.e., $\tilde{\Theta}_M^c$, using the appropriate Bayesian inference method from Section 3.
3: Solve (8). Let $K_\text{cl}$ denote the optimal solution of (8). Set $K^{(0)} \leftarrow \infty$, $K^{(1)} \leftarrow K_\text{cl}$ and $k \leftarrow 1$.
4: **while** $|J_M^c(K^{(k)}) - J_M^c(K^{(k-1)})| > \epsilon$ **do**
5:     Solve $K^* = \arg\min_K \hat{J}_M^c(K, K^{(k)})$. Set $K^{(k+1)} \leftarrow K^*$ and $k \leftarrow k+1$.
6: **end while**
7: **return** $K^{(k)}$ as the control policy.

---

**Remark 4.1.** *This sequential SDP approach can be applied in other robust control settings, e.g., mixed $H_2/H_\infty$ [20], to improve on the common Lyapunov solution, cf. Section 5.1 for an illustration.*

### 4.3 Robustness

Hitherto, we have considered the performance component of the robust control problem, namely minimization of the expected cost; we now address the robust stability requirement. It is desirable for the learned policy to stabilize every model in the confidence region $\Theta^c$; in fact, this is necessary for the cost $J^c(K)$ to be finite. Algorithm 1 ensures stability of each of the $M$ sampled systems from $\tilde{\Theta}_M^c$, which implies that $\phi$ stabilizes the entire region as $M \to \infty$. However, we would like to be able to say something about robustness for finite $M$. To this end, we make two remarks. First, if closed-loop stability of each sampled model is verified with a common Lyapunov function, then the policy stabilizes the convex hull of the sampled systems:

**Theorem 4.2.** *Suppose there exists $K \in \mathbb{R}^{n_x \times n_u}$ such that $(A_i + B_iK)'X(A_i + B_iK) - X \prec 0$ for $X \succ 0$ and all $\Theta = \{A_i, B_i\}_{i=1}^N$. Then $(A + BK)'X(A + BK) - X \prec 0$ for all $\{A, B\} \in \text{conv}\Theta$, where $\text{conv}\Theta$ denotes the convex hull of $\Theta$.*

The proof of Theorem 4.2 is given in A.2.3. The conditions of Theorem 4.2 hold for the common Lyapunov approach in (8), and can be made to hold for Algorithm 1 by introducing an additional Lyapunov stability constraint (with common Lyapunov function) for each sampled system, at the expense of some conservatism. Second, we observe empirically that Algorithm 1 returns policies that very nearly stabilize the entire region $\Theta^c$, despite a very modest number of samples $M$ relative to the dimension of the parameter space, cf. Section 5.1, in particular Figure 2. In principle, results from probabilistic robust control could be used to bound the number of samples required for such robustness properties, cf. e.g., [9, Theorem 1], however, at least for the examples in this paper, such bounds appear to be quite conservative. Furthermore, a number of recent papers have investigated sampling (or grid) based approaches to stability verification of control policies, e.g., [54, 4, 6]. Understanding why policies from Algorithm 1 generalize effectively to the entire region $\Theta^c$, as well as clarifying connections to probabilistic robust control, are interesting topics for future research.

# 5 Experimental results

## 5.1 Numerical simulations using synthetic systems

In this section, we study the infinite horizon LQR problem specified by

$$A_{\text{tr}} = \text{toeplitz}(a, a'), \ a = [1.01, 0.01, 0, \ldots, 0] \in \mathbb{R}^{n_x}, \ B_{\text{tr}} = I, \ \Pi_{\text{tr}} = I, \ Q = 10^{-3}I, \ R = I,$$

where $\text{toeplitz}(r, c)$ denotes the Toeplitz matrix with first row $r$ and first colum $c$. This is the same problem studied in [17, §6] (for $n_x = 3$), where it is noted that such dynamics naturally arise in consensus and distributed averaging problems. To obtain problem data $\mathcal{D}$, each *rollout* involves simulating (1), with the true parameters, for $T = 6$ time steps, excited by $u_t \sim \mathcal{N}(0, I)$ with $x_0 = 0$. Note: to facilitate comparison with [17], we too shall assume that $\Pi_{\text{tr}}$ is known. Furthermore, for all experiments $\Theta^c$ will denote a 95% confidence region, as in [17]. We compare the following methods of control synthesis: **existing methods:** (i) *nominal*: standard LQR using the nominal model from the least squares, i.e., $\{A_{\text{ls}}, B_{\text{ls}}\} := \arg\min_{A,B} \sum_{r=1}^{N} \sum_{t=1}^{T} |x_t^r - Ax_{t-1}^r - Bu_{t-1}^r|^2$; (ii) *worst-case*: optimize for worst-case model (95% confidence) s.t. robust stability constraints, i.e., the method of [17, §5.2]; (iii) $H_2/H_\infty$: enforce stability constraint from [17, §5.2], but optimize performance for the nominal model $\{A_{\text{ls}}, B_{\text{ls}}\}$; **proposed method(s):** (iv) *CL*: the common Lyapunov relaxation of 8; (v) *proposed*: the method proposed in this paper, i.e., Algorithm 1; **additional new methods:** (vi) *alternate-r*: initialize with the $H_2/H_\infty$ solution, and apply the iterative optimization method proposed in Section 4.2, cf. Remark 4.1; (vii) *alternate-s*: optimize for the nominal model $\{A_{\text{ls}}, B_{\text{ls}}\}$, enforce stability for the sampled systems in $\tilde{\Theta}_M^c$. Before proceeding, we wish to emphasize that the different control synthesis methods have different objectives; a lower cost does not mean that the associated method is 'better'. This is particularly true for *worst-case* which seeks to optimize performance for the worst possible model so as to bound the cost on the true system.

To evaluate **performance**, we compare the cost of applying a learned policy $K$ to the true system $\theta_{\text{tr}} = \{A_{\text{tr}}, B_{\text{tr}}\}$, to the optimal cost achieved by the optimal controller $K_{\text{lqr}}$ (designed using $\theta_{\text{tr}}$), i.e., $J(K|\theta_{\text{tr}})/J(K_{\text{lqr}}|\theta_{\text{tr}})$. We refer to this as 'LQR suboptimality.' In Figure 1 we plot LQR suboptimality is shown as a function of the number of rollouts $N$, for $n_x = 3$. We make the following observations. Foremost, the methods that enforce stability 'stochastically' (i.e. point-wise), namely *proposed* and *alternate-s*, attain significantly lower costs than the methods that enforce stability 'robustly'. Furthermore, in situations with very little data, e.g. $N = 5$, the robust control methods are usually unable to find a stabilizing controller, yet the *proposed* method finds a stabilizing controller in the majority of trials. Finally, we note that the iterative procedure in *proposed* (and *alternate-s*) significantly improves on the common-Lyapunov relaxation *CL*; similarly, *alternate-r* consistently improves upon $H_2/H_\infty$ (as expected).

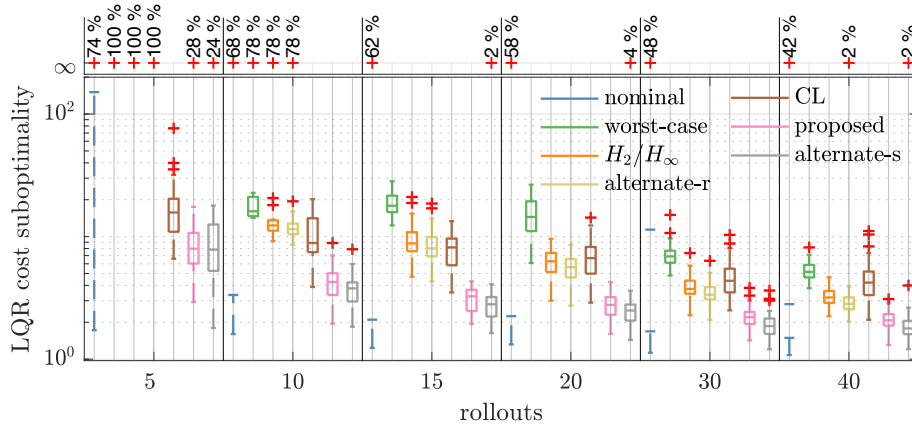

Figure 1: LQR suboptimality as a function of the number of rollouts (i.e. amount of training data). $\infty$ suboptimality denotes cases in which the method was unable to find a stabilizing controller for the *true* system (including infeasibility of the optimization problem for policy synthesis), and the % denotes the frequency with which this occurred for the 50 experimental trials conducted.

Table 1: Median % of unstable closed-loop models, with open-loop models sampled from a 95% confidence region of the posterior, for system of varying dimension $n_x$; cf. Section 5.1 for details. 50 experiments were conducted, with $N = 50$. The policy synthesis optimization problems were always feasible, except for the *worst-case* method, which was infeasible in 46% of trials when $n_x = 12$. $H_2/H_\infty$ and *alternate-r* have the same robustness guarantees as *worst-case*, and are omitted.

| $n_x$ | optimal | nominal | worst-case | CL | **proposed** | alternate-s |
|---|---|---|---|---|---|---|
| 3 | 61.6 | 28.75 | 0 | 0 | 0.10 | 1.35 |
| 6 | 95.37 | 58.41 | 0 | 0 | 0.18 | 1.76 |
| 9 | 99.6 | 81.9 | 0 | 0 | 0.24 | 1.40 |
| 12 | 100 | 94.28 | 0 | 0 | 0.27 | 1.27 |

Table 2: Mean computation times in seconds for control synthesis for system of varying dimension $n_x$; cf. Section 5.1 for details. 50 experiments were conducted, with $N = 50$.

| $n_x$ | worst-case | $H_2/H_\infty$ | CL | **proposed** | alternate-s |
|---|---|---|---|---|---|
| 3 | 1.91 | 0.159 | 0.605 | 20.3 | 4.56 |
| 6 | 2.05 | 0.173 | 0.962 | 28.9 | 13.4 |
| 9 | 2.51 | 0.208 | 1.79 | 48.1 | 27.1 |
| 12 | 3.72 | 0.329 | 3.90 | 96.8 | 62.9 |

It is natural to ask whether the reduction in cost exhibited by *proposed* (and *alternate-s*) come at the expense of **robustness**, namely, the ability to stabilize a large region of the parameter space. Empirical results suggest that this is *not* the case. To investigate this we sample 5000 fresh (i.e. not used for learning) models from $\Theta^c \cap \mathcal{M}$ and check closed-loop stability of each; this is repeated for 50 independent experiments with varying $n_x$ and $N = 50$. The median percentage of models that were *unstable* in closed-loop is recorded in Table 1. We make two observations: (i) the *proposed* method exhibits strong robustness. Even for $n_x = 12$ (i.e., 288-dim parameter space), it stabilizes more than 99% of samples from the confidence region, with only $M = 100$ MC samples. (ii) when the robust methods (*worst-case*, $H_2/H_\infty$, *alternate-r*) are feasible, the resulting policies were found to stabilize 100% of samples; however, for $n_x = 12$, the methods were infeasible almost half the time, whereas *proposed* always returned a policy. Further evidence is provided in Figure 2, which plots robustness and performance as a function of the number of MC samples, $M$. For $n_x = 3$ and $M \geq 800$, the entire confidence region is stabilized with very high probability, suggesting that $M \to \infty$ is not required for robust stability in practice.

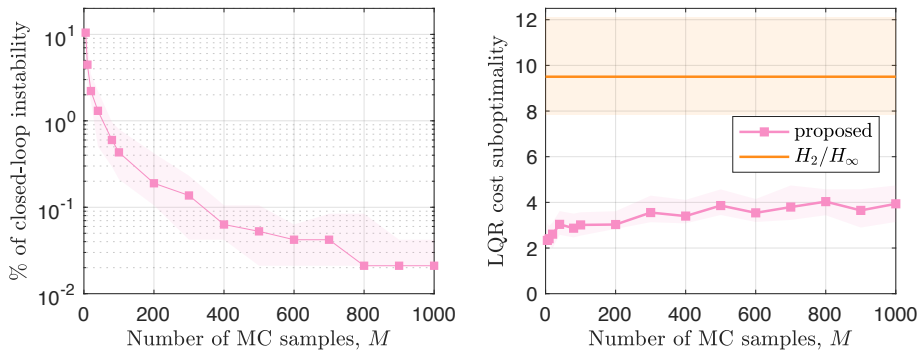

Figure 2: (left) Median % of unstable closed-loop models, with open-loop models sampled from a 95% confidence region of the posterior, for $n_x = 3$ and $N = 15$, as a function of the number of samples $M$ used in the MC approximation (4). (right) LQR suboptimality as a function of $M$. 50 experiments were conducted, cf. Section 5.1 for details. Shaded regions cover the interquartile range.

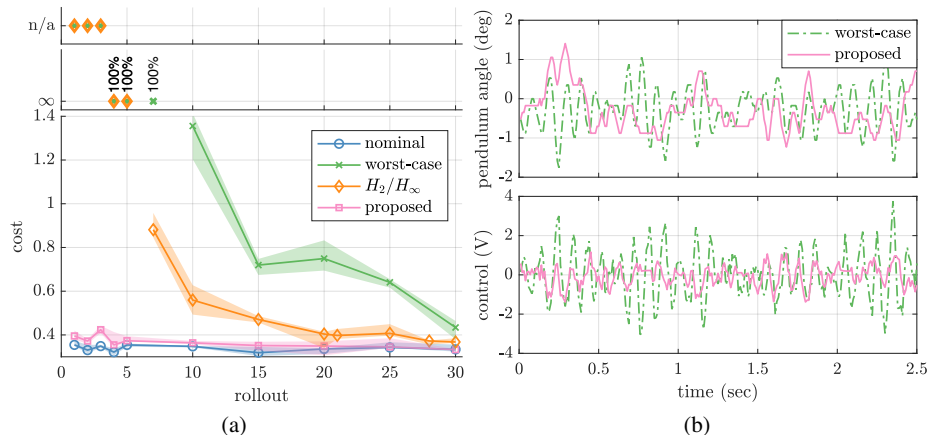

Figure 3: (a) (Median) LQR cost on real-world pendulum experiment, as a function of the number of rollouts. $\infty$ cost denotes controllers that resulted in instability during testing. n/a denotes cases in which the synthesis problem was infeasible. Five trials were conducted to evaluate the cost of each policy. The shaded region spans from the minimum to maximum cost. Note: for this particular experiment, the nominal models from least squares happened to yield stabilizing controllers that offered good performance. Such behavior is not to be expected in general, cf. Figure 1. (b) pendulum angle and control signal recorded after 10 rollouts.

## 5.2 Real-world experiments on a rotary inverted pendulum

We now apply the proposed algorithm to the classic problem of stabilizing a (rotary) inverted pendulum, on real (i.e. physical) hardware (Quanser QUBE 2), cf. A.3 for details. To generate training data, the superposition of a non-stabilizing control signal and a sinusoid of random frequency is applied to the rotary arm motor while the pendulum is inverted. The arm and pendulum angles (along with velocities) are sampled at 100Hz until the pendulum angle exceeds $20°$, which takes no more than 5 seconds. This constitutes one rollout. We applied the *worst-case*, $H_2/H_\infty$, and *proposed* methods to optimize the LQ cost with $Q = I$ and $R = 1$. To generate bounds $\epsilon_A \geq \|A_{ls} - A_{tr}\|_2$ and $\epsilon_B \geq \|B_{ls} - B_{tr}\|_2$ for *worst-case* and $H_2/H_\infty$, we sample $\{A_i, B_i\}_{i=1}^{5000}$ from a 95% confidence region of the posterior, using Gibbs sampling, and take $\epsilon_A = \max_i \|A_{ls} - A_i\|_2$ and $\epsilon_B = \max_i \|B_{ls} - B_i\|_2$. The *proposed* method used 100 such samples for synthesis. We also applied the *least squares policy iteration* method [28], but none of the policies could stabilize the pendulum given the amount of training data. Results are presented in Figure 3, from which we make the following remarks. First, as in Section 5.1, the *proposed* method achieves high performance (low cost), especially in the low data regime where the magnitude of system uncertainty renders the other synthesis methods infeasible. Insight into this performance is offered by Figure 3(b), which indicates that policies from the *proposed* method stabilize the pendulum with control signals of smaller magnitude. Second, performance of the *proposed* method converges after very few rollouts. Data-inefficiency is a well-known limitation of RL; understanding and mitigating this inefficiency is the subject of considerable research [17, 48, 18, 42, 23, 24]. Investigating the role that a Bayesian approach to uncertainty quantification plays in the apparent sample-efficiency of the proposed method is an interesting topic for further inquiry.

## Acknowledgments

This research was financially supported by the Swedish Foundation for Strategic Research (SSF) via the project *ASSEMBLE* (contract number: RIT15-0012) and via the projects *Learning flexible models for nonlinear dynamics* (contract number: 2017-03807) and *NewLEADS - New Directions in Learning Dynamical Systems* (contract number: 621-2016-06079), both funded by the Swedish Research Council.

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
