[Supplementary Material]

# A Supplementary material

## A.1 Sampling from the posterior distribution

### A.1.1 Case I: $\Pi_{\text{tr}}$ known

First, consider the case where $\Pi_{\text{tr}}$ is known; i.e., $\theta = \{A, B\}$. From Bayes' rule, we have

$$\pi(\theta) := p(\theta|\mathcal{D}) = \frac{1}{p(\mathcal{D})} p(\mathcal{D}|\theta)p(\theta) \propto p(\theta) \prod_{i=1}^{N} p(x_+^i | x_-^i, u^i, \theta) =: \bar{\pi}(\theta), \tag{12}$$

where $\mathcal{D} = \{x_+^i, x_-^i, u^i\}_{i=1}^{N}$ and $p(x_+^i | x_-^i, u^i, \theta) = \mathcal{N}\left(x_+^i - Ax_-^i - Bu^i, \Pi\right)$. When $\Pi$ is known, we can express the likelihood $p(\mathcal{D}|\theta)$ in a form equivalent to an (un-normalized) Gaussian distribution over $\theta$, i.e.,

$$\prod_{i=1}^{N} p(x_+^i | x_-^i, u^i, \theta) \tag{13a}$$

$$\propto \exp\left(-\frac{1}{2} \sum_i |x_+^i - Ax_-^i - Bu^i|^2_{\Pi^{-1}}\right) \tag{13b}$$

$$= \exp\left(-\frac{1}{2} \sum_i |x_+^i - D_{x,u}^i \theta_{AB}|^2_{\Pi^{-1}}\right) \tag{13c}$$

$$= \exp\left(-\frac{1}{2} \left(\theta'_{AB}\left(\sum_i D_{x,u}^{i'}\Pi^{-1}D_{x,u}^i\right)\theta_{AB} - 2\theta'_{AB}\sum_i D_{x,u}^{i'}\Pi^{-1}x_+^i + \sum_i x_+^{i'}\Pi^{-1}x_+^i\right)\right) \tag{13d}$$

$$\propto \mathcal{N}\left(\mu, \Sigma\right), \tag{13e}$$

where $\Sigma = \left(\sum_i D_{x,u}^{i'}\Pi^{-1}D_{x,u}^i\right)^{-1}$, $\mu = \Sigma\left(\sum_i D_{x,u}^{i'}\Pi^{-1}x_+^i\right)$, $\theta_{AB} = \text{vec}\left(A'; B'\right)$, and $D_{x,u}^i = I_{n_x} \otimes [x_-^{i'} \ u^{i'}]$. This implies that $\pi(\theta) = \mathcal{N}\left(\mu, \Sigma\right)p(\theta)$. Therefore, when the prior $p(\theta)$ is non-informative ($p(\theta) \propto 1$) or Gaussian (self-conjugate), the posterior is also Gaussian.

### A.1.2 Case II: $\Pi_{\text{tr}}$ unknown

Next, consider the generic case in which all parameters are unknown. Then $\theta = \{A, B, \Pi\}$. One approach to sampling from the posterior involves Gibbs sampling [12], i.e., alternating between the following two sampling steps:

$$\{A_k, B_k\} \sim p(A, B|\Pi_{k-1}, \mathcal{D}), \tag{14}$$

$$\Pi_k \sim p(\Pi|A_k, B_k, \mathcal{D}) \tag{15}$$

to form the Markov Chain $\{A_k, B_k, \Pi_k\}_{k=1}^{\infty}$. As demonstrated in A.1.1, the distribution $p(A, B|\Pi_{k-1}, \mathcal{D})$ is Gaussian, so sampling is straightforward. To sample from $p(\Pi|A_k, B_k, \mathcal{D})$, first note

$$p(\Pi|A, B, \mathcal{D}) \propto p(\mathcal{D}|A, B, \Pi)p(\Pi). \tag{16}$$

Observe that

$$p(\mathcal{D}|A, B, \Pi) \propto \frac{1}{\det(\Pi)^{\frac{N}{2}}} \exp\left(-\frac{1}{2}\sum_i |x_+^i - Ax_-^i - Bu^i|^2_{\Pi^{-1}}\right)$$

$$= \frac{1}{\det(\Pi)^{\frac{N}{2}}} \exp\left(-\frac{1}{2}\text{tr } \Phi\Pi^{-1}\right), \ \Phi := \sum_i (x_+^i - Ax_-^i - Bu^i)(x_+^i - Ax_-^i - Bu^i)'$$

$$\propto \mathcal{W}^{-1}(\Phi, \nu), \nu = N - n_x - 1,$$

where $\mathcal{W}^{-1}(\cdot, \cdot)$ denotes the inverse Wishart distribution. Note, if $N \leq n_x + 1$ then $\nu$ is not valid. However, we may consider a prior $p(\Pi)$ such as $p(\Pi) \propto \det(\Pi)^{-\frac{n_x+1}{2}}$ (Jeffreys' noninformative prior) which means

$$p(\Pi|A, B, \mathcal{D}) \propto \frac{1}{\det(\Pi)^{\frac{N+n_x+1}{2}}} \exp\left(-\frac{1}{2}\text{tr } \Phi\Pi^{-1}\right) \propto \mathcal{W}^{-1}(\Phi, \nu), \tag{17}$$

where $\nu = N > 0$. This is a well-defined inverse Wishart distribution, sampling from which is straightforward.

## A.2 Proofs

### A.2.1 Proof of Lemma 4.1

**Lemma.** $T(S, S_0) \preceq S^{-1}$ for all $S, S_0 \in \mathbb{S}_{++}^n$, where $T(S, S_0)$ denotes the first-order Taylor series expansion of $S^{-1}$ about $S_0$.

*Proof.* Let $D = S - S_0 = L'L$, i.e, $L$ is the Cholesky factorization of $D$. Then,

$$S_0 \succ 0 \iff S_0^{-1} \succ 0 \implies LS_0^{-1}L' \succeq 0 \iff I + LS_0^{-1}L' \succeq I \iff (I + LS_0^{-1}L')^{-1} \preceq I$$
$$\iff S_0^{-1}L'(I + LS_0^{-1}L')^{-1}LS_0^{-1} \preceq S_0^{-1}L'LS_0^{-1}$$
$$\iff S_0^{-1} - S_0^{-1}L'(I + LS_0^{-1}L')^{-1}LS_0^{-1} \succeq S_0^{-1} - S_0^{-1}DS_0^{-1}$$
$$\iff (S_0 + L'L)^{-1} \succeq S_0^{-1} - S_0^{-1}(S - S_0)S_0^{-1} \iff S^{-1} \succeq T(S, S_0),$$

where the penultimate implication invokes the Woodbury matrix inversion identity [37, eq. 159]. □

### A.2.2 Proof of Theorem 4.1

**Theorem.** *Let $\hat{J}_M^c(K, \bar{K})$ be defined as in* (11)*, with $\bar{K}$ such that $J_M^c(\bar{K})$ is finite. Then $\hat{J}_M^c(K, \bar{K})$ is a convex upper bound on $J_M^c(K)$, i.e., $\hat{J}_M^c(K, \bar{K}) \geq J_M^c(K) \,\forall K$. Furthermore, the bound is 'tight' at $\bar{K}$, i.e., $\hat{J}_M^c(\bar{K}, \bar{K}) = J_M^c(\bar{K})$.*

*Proof.* We will prove that $\hat{J}(K, \bar{K}|\theta_i)$ is a tight convex bound on $J(K|\theta_i)$, as this implies that $\hat{J}_M^c(K, \bar{K}) := \frac{1}{M}\sum_i \hat{J}(K, \bar{K}|\theta_i)$ is a tight convex bound on $J_M^c(K) := \frac{1}{M}\sum_i J(K|\theta_i)$. First, we prove convexity. $\hat{J}(K, \bar{K}|\theta_i)$ is defined as the supremum over an infinite family of convex functions over a compact convex set, and is therefore a itself convex function. Note: the minimum of a linear function, e.g. $\min_{X_i \in \mathbb{S}_\epsilon^{n_x}} \operatorname{tr} X_i G_i G_i'$ can be trivially expressed as the supremum of a convex function, i.e., $\sup_{X_i \in \mathbb{S}_\epsilon^{n_x}} -\operatorname{tr} X_i G_i G_i'$. Next, we prove the upper bound. From Lemma 4.1, $X_i^{-1} \succeq T_i(X_i, \bar{X}_i)$ for all $X_i \in \mathbb{S}_\epsilon^{n_x}$. Therefore, any $X_i \in \mathbb{S}_\epsilon^{n_x}$ that satisfies (10b) also satisfies (9b). This means the feasible set of (10) is a subset of the feasible set of (9), hence $\hat{J}(K, \bar{K}|\theta_i) \geq J(K|\theta_i)$. Finally, we prove tightness. As we have already proved $\hat{J}(K, \bar{K}|\theta_i) \geq J(K|\theta_i)$, it suffices to prove that $\bar{X}_i$ is a feasible solution to (10b). As $T(\bar{X}_i, \bar{X}_i) = \bar{X}_i^{-1}$, for $K = \bar{K}$ and $X_i = \bar{X}_i$ (10b) is equivalent to (9b), which is feasible by definition of $\bar{X}_i$. Hence, $\bar{X}_i$ is a feasible solution of (10), that achieves $\operatorname{tr} \bar{X}_i G_i G_i' = J(\bar{K}|\theta_i)$, by definition of $\bar{X}_i$. □

### A.2.3 Proof of Theorem 4.2

**Theorem.** *Suppose there exists $K \in \mathbb{R}^{n_x \times n_u}$ such that $(A_i + B_i K)'X(A_i + B_i K) - X \prec 0$ for $X \succ 0$ and all $\Theta = \{A_i, B_i\}_{i=1}^N$. Then $(A + BK)'X(A + BK) - X \prec 0$ for all $\{A, B\} \in \operatorname{conv}\Theta$, where $\operatorname{conv}\Theta$ denotes the convex hull of $\Theta$.*

*Proof.* It is sufficient to show that $(A + BK)'X(A + BK) - X \prec 0$ defines a convex set in terms of $(A, B)$. By the Schur complement,

$$(A + BK)'X(A + BK) - X \prec 0 \iff \begin{bmatrix} X & (A+BK)' \\ A+BK & X^{-1} \end{bmatrix} \succ 0,$$

which is convex in $A$, $B$ for given (i.e. fixed) $K$ and $X$. □

## A.3 Additional material for experiments on the rotary inverted pendulum

### A.3.1 System description

The Quanser QUBE-Servo 2 inverted pendulum is depicted in Figure 4. The system consists of an actuated arm that rotates in the horizontal plane; actuation is provided by an electrical motor. Attached to the end of the rotary arm is an un-actuated pendulum, which is free to rotate. The voltage applied to the electric motor constitutes the control input $u$ for the LQR problem. Rotary encoders record the angular position of the rotary arm and pendulum, denoted $\alpha_a$ and $\alpha_p$, respectively. These angular positions are fed through a high-pass-filter to provide angular velocity estimates, $(\dot{\alpha}_a, \dot{\alpha}_p)$. The observed state is then given by $x = [\alpha_a, \alpha_p, \dot{\alpha}_a, \dot{\alpha}_p]'$.

### A.3.2 Experimental procedure

To generate one rollout of problem data, we first swing-up the pendulum to the inverted position, stabilized by an LQR designed using a physics-based model of the system. Then we apply the voltage signal $v_t = \bar{K}x_t + \sin(\omega t) + w_t$ and record the resulting state evolution (sampled at 100Hz), until the pendulum angle $|\alpha_p|$ exceeds $20°$, or the rotary arm angle $|\alpha_a|$ exceeds $50°$. Here $\bar{K} = [1, -10, 1, -3]$ constitutes a state feedback policy that does *not* stabilize the system, but does keep the pendulum upright from slightly longer than if it were absent. This extends the typical rollout duration to around 3-5 seconds, before the pendulum angle exceeds $20°$. The angular frequency $\omega$ is randomly sampled each rollout, $\omega \sim \mathcal{U}[20, 35]$, where $\mathcal{U}[a, b]$ denotes the uniform distribution over the interval $[a, b]$. Finally, $w_t$ denotes band-limited white noise, with a sampling time

of $T_s = 0.01$, and a gain of 0.05. $w_t$ represents additional exogenous disturbances that we artificially introduce to the system.

Training data then consists of $\{x_t, u_t\}_{t=0}^T$, where $x_t$ denotes the recorded state sequence, and $u_t = Kx_t + \sin(\omega t)$, i.e., the disturbance $w_t$ is not observed for learning. $T$ is truncated to 500 (i.e. 5 seconds) in the event that the rollout lasts this long. A typical rollout is depicted in Figure 5.

Figure 4: The Quanser QUBE-Servo 2 rotary pendulum, in the inverted position. Photo: www.quanser.com/products/qube-servo-2.

### A.3.3 Bounds for robust methods

The robust synthesis methods *worst-case*, $H_2/H_\infty$, and *alternate-r* require bounds on the error of the least squares estimate, i.e., $\epsilon_A \geq \|A_{\mathrm{ls}} - A_{\mathrm{tr}}\|_2$ and $\epsilon_B \geq \|B_{\mathrm{ls}} - B_{\mathrm{tr}}\|_2$. In [17], these bounds are estimated, with a specific confidence level, via a Boostrap algorithm, assuming that the covariance is known. In our setting, we estimate these bounds as described in Section 5.2, i.e., by sampling from a 95% confidence region of the posterior distribution. This ensures a fair comparison between the methods, as they are, in essence, required to stabilize the same region of the parameter space. We observed, however, that these bounds on the least squares error were too conservative; the magnitude of the uncertainty was too large, and the control synthesis optimization problems were infeasible. The experiments presented in Figure 3 were attained by scaling down these error bounds by a factor of 100. A number of scaling factors were tested, but 100 was found to achieve a reasonable trade-off between robustness and feasibility. It is worth emphasizing that the *proposed* method used the samples from a 95% confidence region without any such scaling.

Figure 5: A typical rollout from the experimental procedure in A.3.2.