[Reviews · NeurIPS 2018]

Reviewer 1



The paper addresses an LQR control problem with unknown dynamics matrices. It is proposed to optimize the expected reward over the posterior distribution of unknown system parameters. The suggested approach first samples system matrices and then discards unstabilizable systems. Using the stabilizable dynamics, an controller is designed that stabilizes all sampled systems and admits a common Lyapunov function. As the controller synthesis problem involves a non-convex matrix inequality constraint, an linearization is proposed to convexify the constraint. This relaxed problem is solved iteratively until convergence. Quality The paper and the theoretical analysis seem to be correct. Mathematical details are provided in sufficient depth. The experimental simulations and the implementation on the inverted pendulum support the author's claims of the applicability and robustness of the method nicely. Clarity The paper is well organized and clearly states the used data and the optimization objective. The authors did a good job in moving mathematical content to the appendix, without breaking the readability of the main paper. Indeed, I enjoyed reading the paper. Originality The proposed approach nicely combines statistical sampling with robust control theory. The main conceptual contribution seems to use the linearization of the matrix inequality to make the robust control approach suitable for the samples system setting. As a whole, the approach is interesting and useful. However, I am missing a comparison to methods studied in the control literature. In particular there seems to be a relation to the field of randomized algorithms for robust control, see e.g., - Roberto Tempo et al. "Randomized Algorithms for Analysis and Control of Uncertain Systems." Springer 2005. This should be carefully evaluated. Significance The approach is clear, theoretically grounded and also appealing for practitioners, as (i) it is fairly simple to implement and (ii) the controllers are guaranteed to be robust (at least in the set of sampled dynamics). The method might find further usage. Minor Comments: - \Pi = GG' is explained only in line 154, but already used in line 119 - Algorithm 1, line 4: Bracket is missing After authors' response: My comments are carefully and convincingly addressed. I recommend the paper for acceptance.

Reviewer 2



Summary: The paper presents a model-based approach for controlling linear systems subject to quadratic costs with stability guarantees. Robust LQ control, yielding a controller K that stabilizes all dynamics (A, B) within an uncertainty set, has previously been studied by Dean et al. This paper attempts to improve robust solutions (make them less conservative) by optimizing over an approximate dynamics posterior, rather than considering the worst case dynamics. The approach is as follows: - Obtain a set of dynamics samples from estimation posterior and restrict it to stabilizable samples. - Formulate the problem as an SDP in the inverse value matrix X^{-1} and controller K, where the objective is a conservative upper bound on the average control cost of all dynamics samples (A_i, B_i, Pi_i). This is the common Lyapunov (CL) solution. - Propose a less conservative convex upper bound as follows. Given an initial controller \bar{K}, find the dynamics sample with the corresponding lowest cost \bar{X}. Then solve a modified problem where each sample gets its own value matrix X_i but K is still shared, and each X_i^{-1} is approximated using a Taylor series around \bar{X}. This procedure can be applied iteratively to improve CL or other solutions. - The final K stabilizes all dynamics in the convex hull of the samples for the CL approach, and also for the relaxed (proposed) approach if an additional stability constraint is introduced (but I don’t think it is). - Experiments are run on small-scale synthetic dynamics similar to previous work. Data is obtained by exciting the system with Gaussian inputs and force-resetting to 0 every 6 time steps. The proposed algorithm (and variations) do seem to find a good tradeoff between stability and cost; while they occasionally yield unstable controllers, their cost is lower than that of the fully robust controllers. Experiments are also run on inverted pendulum with similar conclusions. Quality & clarity. This is a high-quality paper with clear writing, explanations, and derivations. It proposes an interesting approach to robust control of LQ systems with unknown dynamics which trades occasional instability for lower control costs. Originality: The contributions are mostly extensions of previous work, though the new upper bound is original. Wrt the SDP formulation: at the beginning of Section 4, authors state that the principal contribution (btw principle-->principal lines 138, 145) of their paper is an SDP formulation of the problem. However, SDP formulations of LQ control have been known for a while, and the paper should include at least one reference. e.g. - V. Balakrishnan and L. Vandenberghe. Semidefinite programming duality and linear time-invariant systems. IEEE Transactions on Automatic Control, 2003. - A Cohen et al. Online linear quadratic control with adversarially changing costs, ICML 2018. (most recently) Standard SDP formulations just solve for X and subsequently obtain K= −(R + B’XB)^{−1}B’XA. The formulation in the paper is only different in that it also optimizes over K in order to tie multiple X_i using the same K. Significance: While the results are nice (improve overly conservative solutions while mostly maintaining stability), this approach may not be too useful for practical problems due to increased computation reqiurements. Solving LQ control using SDP relaxations is practical only for small-scale systems. In this paper, the computation cost is amplified by the number of dynamics samples and the use of an iterative procedure. This is perhaps not an issue in this paper since the controllers are computed offline from force-reset episodes. But it limits the use of the developed technique in large-scale problems, as well as safety- and stability-critical adaptive settings. A few additional questions about experiments: - What happens if you make your approach fully stable - is the cost substantially higher? - The nominal controller seems to find lowest-cost solutions (when stable). Why is it not included in the pendulum experiments?

Reviewer 3



This paper presents a well thoughtout procedure for estimating the system parameters of a LQR system identification problem from data. Unlike H-inf, which assumes as worst-case disturbace model, the paper formulates as solution based on a confidence region in parameter space from data using Bayes rule. The formulation leads to a non-convex optimization problem. The proposed algorithm iteratively minimizes a convex upper-bound converging to a local minimum. The problem that the paper aims to tackle, safe/robust RL, is important and compelling. The proposed approach is motivated by deficiencies in prior work and its development, at a high level, is easy to follow. The second set of experiments, on a real dynamical system, build confidence in its effectiveness. The approach is solid and is relevant to the NIPS audience. Below are some concerns regarding clarity and significance. Completely missing from the paper is a discussion of computational complexity. Solving an SDP with an off-the-shelf solver costs n^3.5*log(1/epsilon) where and n is usually the square of the problem dimension. The initialization SDP (8), grows with the number of samples and should quickly become computationally infeasible. The iterative procedure at least decomposes into M separate SDPs. Even an informal discussion about the observed computational requirements, e.g., how restarting the solver from iteration to iteration would be appreciated, and how it compares to the alternatives. The statistical complexity of the approach is discussed experimentally (both in respect to the MCMC sampler as well as in comparison to prior work), but a more formal discussion would greatly improve the paper. Particularly since the computational complexity is dependent on the number of samples. The interplay between the mixing time of the MCMC and the rejection of unstabilizable systems is also important. It is hard to properly evaluate the significance, i.e., the applicability, of the approach without these complexity results. Unfortunately, my guess is that, like most SDP relaxations, it will only be of academic interest. Though the development of the approach is intuitive, the details of the paper can be hard to follow. This will turn away readers and hinder reproducability. The notation is challenging: - pi is probably unnecessary and different than Pi, just use p - consider elimating theta, e.g., (6b) A_i,B_i,\Pi_i = theta_i not immediate or make it explicit like (2) - M, theta_i not defined after first use in (4) until next section - c is both overloaded, used in both Theta^c and as an upper-bound in S^n_epsilon (whose dependence on c is not part of the notation) - J^c_M is not an MC approximation to J^c since the \bar S is now included - \bar S and S are not related - J and \hat J's dependence on i is not obvious (9a-b,10a-b) Minor comments: - Theta^c is "the c% confidence interval", can't there be more than one? - LMI not defined (line 148) - remove (0)'s from table 1 as only one entry differs